# Urinary Acidification Does Not Explain the Absence of Nephrocalcinosis in a Mouse Model of Familial Hypomagnesaemia with Hypercalciuria and Nephrocalcinosis (FHHNC)

**DOI:** 10.3390/ijms25031779

**Published:** 2024-02-01

**Authors:** Amr Al-Shebel, Geert Michel, Tilman Breiderhoff, Dominik Müller

**Affiliations:** 1Charité–Universitätsmedizin Berlin, Corporate Member of Freie Universität Berlin and Humboldt-Universität zu Berlin, Department of Pediatric Gastroenterology, Nephrology and Metabolic Diseases, Augustenburger Platz 1, 13353 Berlin, Germany; tilman.breiderhoff@charite.de (T.B.); dominik.mueller@charite.de (D.M.); 2Charité–Universitätsmedizin Berlin, Corporate Member of Freie Universität Berlin and Humboldt-Universität zu Berlin, Research Institutes for Experimental Medicine, Transgenic Technologies, Robert Rössle Str. 10, 13125 Berlin, Germany; geert.michel@charite.de

**Keywords:** familial hypomagnesaemia with hypercalciuria and nephrocalcinosis (FHHNC), nephrocalcinosis, urinary acidification, claudin-16, Atp6v1b1

## Abstract

Patients with mutations in *Cldn16* suffer from familial hypomagnesaemia with hypercalciuria and nephrocalcinosis (FHHNC) which can lead to renal insufficiency. Mice lacking claudin-16 show hypomagnesemia and hypercalciuria, but no nephrocalcinosis. Calcium oxalate and calcium phosphate are the most common insoluble calcium salts that accumulate in the kidney in the case of nephrocalcinosis, however, the formation of these salts is less favored in acidic conditions. Therefore, urine acidification has been suggested to limit the formation of calcium deposits in the kidney. Assuming that urine acidification is causative for the absence of nephrocalcinosis in the claudin-16-deficient mouse model, we aimed to alkalinize the urine of these mice by the ablation of the subunit B1 of the vesicular ATPase in addition to claudin-16. In spite of an increased urinary pH in mice lacking claudin-16 and the B1 subunit, nephrocalcinosis did not develop. Thus, urinary acidification is not the only factor preventing nephrocalcinosis in claudin-16 deficient mice.

## 1. Introduction

The kidneys play an essential role in maintaining the homeostasis of salts, minerals and water, in addition to waste excretion. This organ consists of about a million units called nephrons, of which each nephron has a glomerulus and a tubule. The glomerulus functions as a filter. Plasma components, except for large proteins and cells, exit the blood and enter the glomerular filtrate by passing through the glomerular filtration barrier. Glomerular filtrate contains the required amount of important salts, minerals and water, in addition to waste molecules that should be excreted. By flowing through the nephron tubule, the consistency of the filtrate changes as needed ingredients are reabsorbed into the blood while waste and extra amounts remain in the nephron to form the urine. To perform its function, the nephron tubule has different segments that possess different cell types expressing different transporters, channels and tight junction proteins.

Epithelial transport in the renal tubules takes the transcellular or paracellular route. The former requires energy while the latter is passive. Tight junctions are supramolecular organizations of membrane proteins with their adapter and scaffolding proteins that form intercellular barrier between epithelial cells [1]. Claudins are the major constituent of the tight junction. These membrane proteins interact with each other within the same cell and in the neighboring cells, and these interactions are important in order to form a tight barrier or pores between two cells. The claudins define the characteristics of the paracellular epithelial transport. Different claudins are expressed along the nephron to regulate the permeability and selectivity of the paracellular pathway in every tubular segment. The importance of claudin-10a, claudin-10b, claudin-2, claudin-12, claudin-16, claudin-19 and claudin-14 in the paracellular reabsorption of Na^+^, Cl^−^, Ca^2+^ and Mg^2+^ has been demonstrated previously [2].

Mutations in *Cldn16* or *Cldn19* that encode claudin-16 and claudin-19, respectively, cause familial hypomagnesemia with hypercalciuria and nephrocalcinosis (FHHNC) (OMIM: 248250), which is an autosomal-recessive renal tubular disorder which sees nephrocalcinosis as a main symptom [3,4,5]. In the thick ascending limb of the loop of Henle (TAL), claudin-16 and claudin-19 facilitate paracellular reabsorption of Ca^2+^ and Mg^2+^. A proportion of ~25% and ~60%, respectively, of the freely filtered cations are reabsorbed in this part of the nephron [6,7,8]. Accordingly, patients with FHHNC suffer from defects in Ca^2+^ and Mg^2+^ reabsorption in TAL with insufficient compensation in the distal convoluted tubules (DCT)—the next nephron segment—where 5–10% of Ca^2+^ and Mg^2+^ are reabsorbed transcellularly via *Trpv5* and *Trpm6* transporters [9,10].

In FHHNC patients, the urinary loss of Ca^2+^ and Mg^2+^ can be accompanied by recurrent urinary tract infections, nephrolithiasis, polyuria, polydipsia, failure to thrive, progressive nephrocalcinosis and eventually chronic renal insufficiency or even end-stage renal disease. Diagnosis with FHHNC usually occurs in early childhood or before adolescence. Currently, the available treatment is symptomatic including oral magnesium supplementation and thiazide diuretics. However, these treatments have no significant effect on the progression of the renal dysfunction. Ultimately, the condition of the patients requires dialysis and kidney transplantation [5].

One of the serious symptoms of FHHNC is nephrocalcinosis, which is characterized by the precipitation of calcium salts in the kidney [11]. Nephrocalcinosis is often caused by defects in renal epithelial transport including mutations in *Cldn16*, *Cldn19*, *Clcn5* and *Slc34A1* [3,4,12,13]. The etiology of calcium deposit formation in the kidney and urinary tract is multifactorial [11]. Hypercalciuria is one of the most important risk factors for nephrocalcinosis formation. FHHNC, Dent’s disease, autosomal dominant hypocalcemia with hypercalciuria (ADHH), Bartter syndromes (I, II and V), distal renal tubule acidosis (dRTA) and primary hyperparathyroidism are examples of pathologies where hypercalciuria and nephrocalcinosis are associated. However, hypercalciuria is not a prerequisite for nephrocalcinosis. For instance, some patients with Dent’s disease were reported to have nephrocalcinosis without hypercalciuria while others have both symptoms [14]. In addition, primary hyperoxaluria leads to nephrocalcinosis without any association with hypercalciuria [11].

In addition, mouse models do not always reproduce the state of nephrocalcinosis observed in humans, indicating the complexity of this pathology [15]. By deleting the *Cldn16* gene, we have generated a FHHNC mouse model displaying hypomagnesemia and hypercalciuria comparable to the patient situation [16]. However, despite the massive renal loss of Ca^2+^, nephrocalcinosis was not detected in young animals nor in adult mice, irrespective of their gender. This finding shows that other unknown mechanisms influence nephrocalcinosis and stone formation.

As such, urinary acidification was described as a part of renal adaptation mechanisms to prevent calcium salt precipitation in the presence of hypercalciuria [17]. The transient receptor potential vanilloid member 5 (TRPV5) channel, encoded by *Trpv5* gene, confers the apical entry step of transcellular Ca^2+^ reabsorption in DCT (where a significant amount of Ca^2+^ is reabsorbed). Likewise, *Trpv5* knockout (KO) mice display high amounts of renal Ca^2+^ wasting [18]. However, nephrocalcinosis was not detected in *Trpv5* KO mice. The urine obtained from *Trpv5* KO mice was significantly more acidic in comparison to that of the control and neutralizing this urine resulted in crystal formation, which showed the importance of low urinary pH in preventing calcium crystal formation. It was shown that these mice had an increased expression of *Atp6v1b1* gene encoding the B1 subunit of the H^+^-ATPase, which secretes protons into the urine and ensures urinary acidification [17]. The additional ablation of the B1 subunit of the H^+^-ATPase resulted in nephrocalcinosis, renal insufficiency, bone loss and poor growth in *Trpv5 Atp6v1b1* dKO mouse model [17]. Thus, urine acidification was shown to be an important factor preventing the formation of calcium precipitants and thus, nephrocalcinosis in *Trpv5* KOs.

In the present study, we tested the hypothesis that urine acidification by vesicular H^+^ ATPase prevents nephrocalcinosis in claudin-16-deficient mice. For this purpose, we deleted the *Atp6v1b1* gene in mice lacking the *Cldn16* gene.

## 2. Results

### 2.1. Deletion of Atp6v1b1 in Claudin-16 Deficient Animals

CRISPR/Cas9-mediated genomic editing was utilized to delete the gene *Atp6v1b1* in *Cldn16* KO mice to obtain mice with simultaneous deletion of *Cldn16* and *Atp6v1b1*. The approach included the design of two guide RNAs, specifically targeting sequences situated both upstream and downstream of the *Atp6v1b1* gene. These guide RNAs were used to direct Cas9 endonuclease to make double strand cleavage in the two target sites. In its turn, the cellular repair mechanism of non-homologous end-joining (NHEJ) was employed, resulting in the creation of an allele characterized by the deletion of the targeted *Atp6v1b1* gene (Figure 1). After identification of mice containing the deleted allele, these were bred to *Cldn16* KO animals to generate claudin-16-deficient animals heterozygous for the deleted *Atp6v1b1* allele. These were then bred to each other to generate claudin-16-deficient animals homozygous for the deleted *Atp6v1b1* allele (double-knockout, dKO). *Cldn16 Atp6v1b1* dKO animals were viable and fertile and were indistinguishable from their littermates as they were only lacking claudin-16.

Immunofluorescence staining of H^+^-ATPase B1 showed the total absence of this protein in kidney sections of *Cldn16 Atp6v1b1* dKO in contrast to the normal staining in kidney sections obtained from *Cldn16* KO mice (Figure 2a). In addition, RT-qPCR analysis showed the lack of *Atp6v1b1* transcripts in *Cldn16 Atp6v1b1* dKO mouse kidneys compared to *Cldn16* KO mouse kidneys (Figure 2b).

### 2.2. Absence of H^+^-ATPase B1 Subunit Resulted in Increased Urinary pH

In order to determine urinary pH upon H^+^-ATPase B1 subunit ablation, spot urine samples were collected from *Cldn16* KO and *Cldn16 Atp6v1b1* dKO. Urinary pH increased in *Cldn16 Atp6v1b1* dKO male mice (7.04 ± 0.03) when compared to *Cldn16* KO male mice (5.30 ± 0.12). A similar effect was detected in female mice where urinary pH increased in *Cldn16 Atp6v1b1* dKOs (6.56 ± 0.15) when compared to *Cldn16* KOs (5.51 ± 0.07) (Figure 3).

### 2.3. Absence of Nephrocalcinosis in Cldn16 KO and Cldn16 Atp6v1b1 dKO Mice

To investigate if urine alkalinization enhances the precipitation of calcium salts in *Cldn16 Atp6v1b1* dKO, kidney sections were stained for nephrocalcinosis. Von Kossa staining and Alizarin Red S staining showed the absence of nephrocalcinosis in *Cldn16* KO male and in *Cldn16 Atp6v1b1* dKO counterpart mice (Figure 4). The same results were obtained in female mice.

## 3. Discussion

In this study, we demonstrate that urine acidification is not the main reason behind the absence of nephrocalcinosis in the *Cldn16* KO mouse model. This conclusion is based on the absence of nephrocalcinosis despite the urine alkalinization induced by the additional deletion of the *Atp6v1b1* gene in the *Cldn16 Atp6v1b1* dKO model.

Calcium and acid excretion in *Trpv5* KO and *Cldn16* KO models was comparable. *Trpv5* KO and *Cldn16* KO mouse models exhibited a comparable increase in calcium urinary excretion of 13-fold and 11-fold, respectively. In addition, comparable urinary acidification was measured in both models (*Trpv5* KO: pH = 5.5, *Cldn16* KO: pH = 5.4 (average of males and females)). Additional ablation of the H^+^-ATPase B subunit resulted in an increase in the urinary pH of the corresponding double-knockout models (*Trpv5 Atp6v1b1* dKO: pH = 7.5, *Cldn16 Atp6v1b1* dKO: pH = 6.8 (average of males and females)). In contrast to *Trpv5 Atp6v1b1* dKO, in our *Cldn16 Atp6v1b1* dKO the alkalinization in the urine did not lead to nephrocalcinosis [16,17].

There are differences between the *Trpv5* KO model and our *Cldn16* KO model in the context of nephrocalcinosis. Claudin-16 acts on the reabsorption of Ca^2+^ and Mg^2+^, whereas *Trpv5* transports Ca^2+^ only. Likewise, whereas *Trpv5* KO mice showed hypercalciuria only, *Cldn16* KO mice showed hypercalciuria accompanied by hypermagnesuria [16,18]. Mg^2+^ has been proposed as a crystal inhibitor that can compete with Ca^2+^ to form more soluble salts. In addition, some studies showed the correlation between Mg^2+^ levels and kidney stones occurrence [20,21,22]. Nevertheless, when considering Mg^2+^ to influence calcium crystal formation, it is of notice that FHHNC patients develop nephrocalcinosis despite the higher concentration of magnesium in the urine [5].

The response of different hormones involved in calcium homeostasis is different between *Trpv5* KO and *Cldn16* KO. In *Trpv5* KO mice, only 1,25-dihydroxyvitamin D3 level was elevated with normal parathyroid hormone (PTH) level in the blood [18], while in *Cldn16* KO mice, both 1,25-dihydroxyvitamin D3 and PTH were significantly elevated [16]. We speculate that the impact of claudin-16 ablation is higher than the impact of *Trpv5* ablation on calcium homeostasis. The expression site along the nephron of both proteins is different. Whereas claudin-16 is expressed in the TAL, *Trpv5* is found in the DCT. In TAL, ~25% of calcium is reabsorbed while only ~5–10% of calcium is reabsorbed in DCT [6,7,8,9,10]. Accordingly, *Cldn16* KO mice have a higher loss of calcium resulting from the loss of claudin-16 in TAL. However, this loss is partially compensated in DCT (the following part of the nephron) by the upregulation of different proteins involved in calcium reabsorption [16]. In contrast to that, the calcium loss in *Trpv5* KO mice originates from the DCT and is not compensated in the latter parts of the nephron, which have no calcium reabsorption mechanisms known. This higher impact of claudin-16 ablation can result in the reported elevation of PTH and 1,25-dihydroxyvitamin D3, while the ablation of *Trpv5* results in the elevation of 1,25-dihydroxyvitamin D3 without PTH elevation [16,18]. 1,25-dihydroxyvitamin D3 synthesis is located in the proximal tubule and can be subject to local fine tuning with a faster response to smaller changes. The calcium sensing receptor (CaSR), expressed in the proximal tubule, might have an effect on the synthesis of 1,25-dihydroxyvitamin D3 by regulating 25-hydroxyvitamin D 1α-hydroxylase, as was shown in vitro [23,24].

The difference is hormonal response between *Cldn16* KO and *Trpv5* KO mouse models can account for different activation levels of CaSR in the collecting duct, therefore resulting in the different responses to hypercalciuria. PTH and 1,25-dihydroxyvitamin D3 can theoretically affect CaSR function. Both the parathyroid hormone receptor and CaSR transcripts were found to co-express at glomeruli, proximal convoluted tubule, proximal straight tubule, cortical thick ascending limb, distal convoluted tubule and cortical collecting duct, which suggests an interaction between these two receptors involved in calcium regulation [25]. However, PTH decreased the expression of CaSR only in the proximal tubule in vivo with no evidence on the function in that study [26]. The vitamin D receptor (VDR) was also shown with immunofluorescence staining to be weakly expressed in the collecting ducts as well as CaSR [27,28]. Since CaSR is suggested to promote urine acidification and diuretic effects in response to hypercalciuria in *Trpv5* KO mice, changes in the activation level of this receptor due to changes in hormonal response can lead to different overall responses in *Cldn16* KO mice.

The difference in phenotype between *Trpv5 Atp6v1b1* dKO and *Cldn16 Atp6v1b1* dKO extends beyond nephrocalcinosis. *Trpv5 Atp6v1b1* dKO mice showed retarded growth directly after birth, 80% of these mice died within 6 weeks after birth and bilateral hydronephrosis in addition to abnormal dilation of the collecting ducts was observed [17]. In contrast, *Cldn16 Atp6v1b1* dKO showed normal appearance and life span. Importantly, nephrocalcinosis was absent when they were sacrificed at the age of 40–44 weeks. The injury caused by nephrocalcinosis cannot explain the complex phenotype of *Trpv5 Atp6v1b1* dKOs, especially since other mouse models with nephrocalcinosis (*Cldn2* KO and *Cldn10* ksp cKO) do not develop this severe pathology [19,29]. More investigations are needed to understand the difference between *Trpv5 Atp6v1b1* dKO and *Cldn16 Atp6v1b1* dKO models.

It is important to mention the difference in nephrocalcinosis between *Atp6v1b1* KO mice and patients with *ATP6V1B1* mutations. Human patients with mutations in *ATP6V1B1* develop distal renal tubule acidosis (dRTA) and suffer from nephrocalcinosis [30], while mice with *Atp6v1b1* deletion show urine alkalinization with no systemic acidosis or nephrocalcinosis. After dietary acid load, *Atp6v1b1* KO mice developed metabolic acidosis. The difference in diets of mice and humans can explain the absence of systemic acidosis in *Atp6v1b1* KO mice [31,32]. As a normal diet was provided to mice in our study, *Atp6v1b1* deletion should not result in metabolic acidosis.

In this study, two types of nephrocalcinosis staining were employed. The first was von Kossa staining, in which silver reacts with calcium phosphate to form a black precipitant. The second was Alizarin Red S staining, in which Alizarin Red S and Ca^2+^ ions precipitate to form brick-red deposits. Combined, the results of these staining methods can confirm the presence of calcium phosphate deposits in the tissue. However, changes in the content of calcium salts in the renal tissues that are not big enough to form deposits are not detected in this method. This method was used to detect calcium deposits in other mouse models with nephrocalcinosis [17,19,29]. In our study, von Kossa staining and Alizarin Red S staining in *Cldn16 Apt6v1b1* dKO kidney sections lacked any positive staining that could be qualitatively comparable to the staining observed in the included positive control or any other positive staining in the literature.

Taken together, nephrocalcinosis happens when calcium salts precipitate in the kidney tissues. However, hypercalciuria is not the only cause of this pathology. There is a difference in the occurrence of nephrocalcinosis between patients and the corresponding animal models [15,16,19]. Urine alkalinization, caused by the deletion of *Atp6v1b1,* was shown to be important in inducing nephrocalcinosis in hypercalciuric mice lacking transient receptor potential vanilloid 5 (TRPV5) [17]. By showing that urine alkalinization does not induce nephrocalcinosis in hypercalciuric mice lacking claudin-16 due to the deletion of *Atp6v1b1*, we emphasize the multifactorial nature of this pathology. Understanding the mechanisms underlying nephrocalcinosis or its preventing factors can be essential in providing improved therapies to nephrocalcinosis patients.

## 4. Materials and Methods

### 4.1. Atp6v1b1 KO Allele Generation

CRISPR/Cas9-mediated genomic editing was used to delete the entire *Atp6v1b1* gene in murine zygotes that were implanted in foster mothers. crRNA-5 (GAAGTCTTAGAGCTACCACC) was designed to anneal upstream to the start codon while crRNA-3 (GGAGAATGGAAGCGGCGAGG) was designed to anneal downstream to the stop codon. crRNA/tracrRNA complexes were annealed by mixing equimolar amounts of crRNAs and tracrRNA. These were heated to 95 °C for 5 min and then slowly cooled to room temperature. Ribonucleoprotein particles (RNPs) were assembled by mixing cRNA/tracrRNA complexes and HiFi Cas9 Protein V3 (Integrated DNA Technologies, Coralville, IA, USA) to reach a final concentration of 2.4 µM (each gRNA) and 4 µM, respectively. The mixture was left at room temperature for 10 min. RNPs were electroporated into murine zygotes in 1 mm cuvettes by applying 30 V in 2 pulses of 3 ms with intervals of 100 ms in a Gene Pulser II (Biorad, Hercules, CA, USA) [33]. Generation of the mouse model was approved by the Landesamt für Gesundheit und Soziales (Lageso) under G0177/17 Amendment E10.

### 4.2. Animal Housing

Mice were housed under standardized conditions (12 h light/dark cycle; 22–24 °C temperature; 55% ± 15% humidity; ad libitum access to standard diet and water). Mice used in the experiment were 40–44 weeks old.

### 4.3. Nephrocalcinosis Staining

Kidneys were collected after sacrificing the mice with cervical dislocation. Kidney tissues were subsequently fixated overnight with phosphate-buffered saline (PBS) solution containing 4% paraformaldehyde (PFA). Then, tissues were dehydrated by incubation for 2–3 h at 4 °C in graded solutions of water to denatured ethanol (water, 50%, 70%, 80%, 95%, 99%). The dehydrated tissues were incubated later in xylene substitute (Neo-Clear™) for 2–3 h at 4 °C. Then, kidneys were incubated in melted paraffin for 2–3 h at 60 °C. Finally, kidney tissues were embedded in paraffin blocks.

Thin sections (5 µm) were cut from the paraffin-embedded kidneys and deparaffinized by two consecutive incubations in Neo-Clear for 5 min. Neo-Clear was then removed, and tissues were rehydrated by incubation in graded solutions of denatured ethanol to water (99%, 95%, 80%, 70%, 50%, water) at room temperature for 5 min each.

For Alizarin Red S staining, kidney sections were submerged for 30 s in Alizarin Red S solution (2% Alizarin Red S, pH 4). Afterwards, sections were dehydrated and mounted.

For von Kossa staining, sections were submerged in 1% aqueous silver nitrate solution and exposed to UV light for 20 min. Then, unreacted silver was removed by incubation in 5% sodium thiosulfate for 5 min. Afterwards, sections were counterstained with nuclear fast red for 5 min. Finally, sections were dehydrated, mounted and visualized under a light microscope.

### 4.4. Spot Urine Collection and pH Measurement

To collect urine, mice were placed in a clean plastic box with a cold bottom. Urine was collected with a pipette. Urinary pH was measured immediately after collection using a thin pH electrode.

### 4.5. Immunofluorescence Staining

Sections with 5 µm thickness were prepared, deparaffinized and rehydrated. Sections were submerged in retrieval buffer (10 mM Tris Base, 1 mM EDTA Solution, 0.05% Tween-20, pH 9.0) in plastic Coplin jars with lids and heated in a pressure cooker for 20 min. Afterwords, sections were washed with PBS buffer containing 0.05% Tween-20 and incubated with a blocking solution (PBS, 10% donkey serum, 0.3% Triton X-100) at room temperature for 1 h. Then, sections were incubated overnight with primary antibodies (1:1000 rabbit-anti-H^+^-ATPase B1) diluted in (PBS, 5% donkey serum, 0.15% Triton X-100) at 4 °C. Afterwards, sections were washed and incubated with donkey anti-rabbit secondary antibody labeled with Alexa Fluor™ Plus 488 (Invitrogen, Waltham, MA, USA) at room temperature for 2 h. Finally, sections were mounted with ProLong™ Diamond Antifade Mountant with DAPI (Invitrogen) and visualized under an epifluorescence microscope. Controls, in which the primary antibody was omitted, were used to subtract the background in the ImageJ program.

### 4.6. Quantitative Reverse Transcription PCR

The extraction of total RNA from kidney tissues was conducted using TRIzol reagent (Life Technologies, Carlsbad, CA, USA). Whole kidney samples were added into centrifuge tubes containing 1ml TRIzol for every 100 mg tissue. Samples were homogenized for 20–30 s at room temperature. Homogenates were incubated for 5 min at room temperature. An amount of 0.2 mL chloroform per 1 mL TRIzol was added and the mixture was vortexed for 15 s, where it was then incubated at room temperature for 5 min. After centrifugation at 12,000× *g* for 10 min, the upper aqueous phase was transferred to a new tube. The aqueous phase contains the isolated RNA and RNeasy Mini Kit (Qiagen, Hilden, North Rhine-Westphalia, Germany) was used for further steps of RNA purification. An amount of 70% ethanol was added to the separated aqueous phase in 1:1 ratio to precipitate the RNA. The mixture was loaded into RNeasy mini column to bind the RNA. After washing steps, DNase I solution was added to the column to digest any genomic DNA remnants. Finally, total RNA was eluted with 30 µL of elution buffer. The first strand of cDNA was synthesized using high-capacity cDNA Reverse Transcription Kit (Applied Biosystems, Waltham, MA, USA). For every sample, 2 µg purified RNA was used to synthesis single-stranded cDNA. cDNA was amplified using primers: *Atp6v1b1*-forward (CCAGAAAGGACCACGGAGATGT), *Atp6v1b1*-reverse (AGGAACTCCAGGTAGAGCAGGT), Tbp-forward (CTTCGTGCAAGAAATGCTGA) and *Tbp*-reverse (TCACTCTTGGCTCCTGTGC). qPCR reaction was performed using PowerUp SYBR Green Master Mix (Applied Biosystems) and samples were processed in technical duplicates. SYBR Green signal was measured in QuantStudio™ 3 System. Expression of genes was normalized per sample to the expression of *Tbp* gene and normalized relative expression was calculated as previously [34]. Mean Cq values of the technical duplicates were calculated and then the general mean of all mean Cq values of all the samples within the control group was obtained. ΔCq per sample per target was calculated by subtracting the general mean Cq of the control group from the mean Cq of samples. ΔCq values were raised to the power of 2 to obtain relative quantity. Normalized expression per sample was calculated by dividing the relative quantity of genes of interest by the relative quantity of the housekeeping gene for every sample.

### 4.7. Statistical Analysis

Welsh’s *t*-test was used. *p*-value < 0.05 was considered to refuse null hypotheses. Figure generation and statistical analysis were performed using GraphPad Prism.

## Figures and Tables

**Figure 1 ijms-25-01779-f001:**
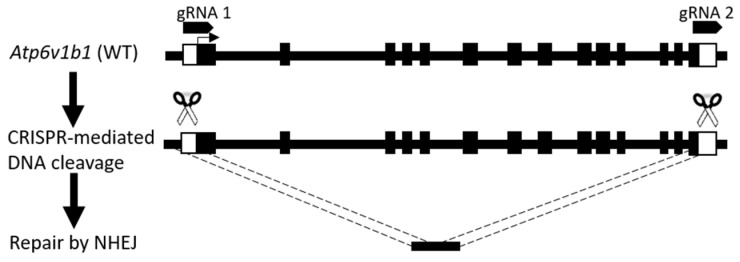
Strategy for the deletion of *Atp6v1b1* gene by CRISPR/Cas9 mediated genomic editing in murine zygotes. Two guide crRNAs were designed corresponding to target sequences upstream and downstream of *Atp6v1b1* gene. Non-homologous end-joining after CRISPR/Cas9 mediated cleavage resulted in an allele with deleted gene.

**Figure 2 ijms-25-01779-f002:**
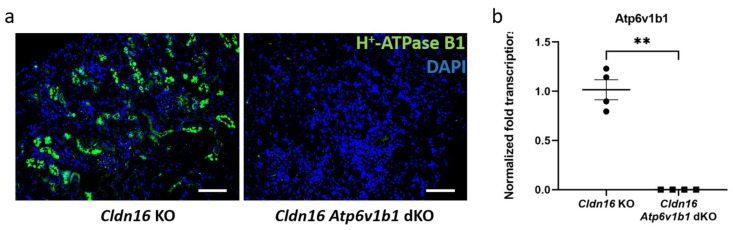
Absence of the H^+^-ATPase B1 subunit in kidneys of *Cldn16 Atp6v1b1* dKO mice. (**a**) Immunofluorescence staining of H^+^-ATPase B1 in *Cldn16* KO (left) and *Cldn16 Atp6v1b1* dKO (right) kidney sections; (**b**) RT-qPCR measurement of the transcription level of *Atp6v1b1* gene in *Cldn16* KO and *Cldn16 Atp6v1b1* dKO kidney tissues; scale bar: 100 µm; n = 4–5; asterisks indicate statistically significant changes ** *p* < 0.01.

**Figure 3 ijms-25-01779-f003:**
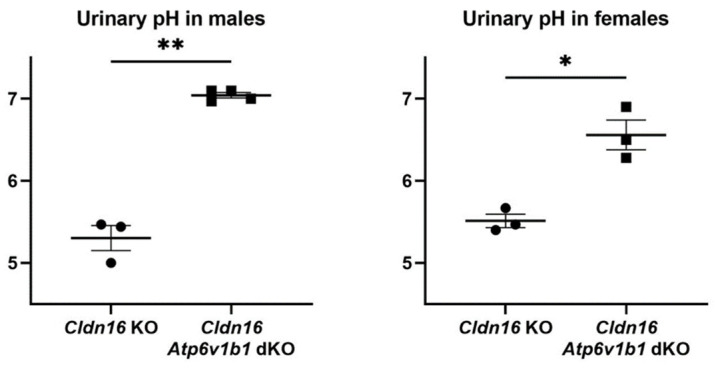
Urinary pH of *Cldn16* KO and *Cldn16 Atp6v1b1* dKO male and female mice. Asterisks indicate statistically significant changes * *p* < 0.05, ** *p* <0.01.

**Figure 4 ijms-25-01779-f004:**
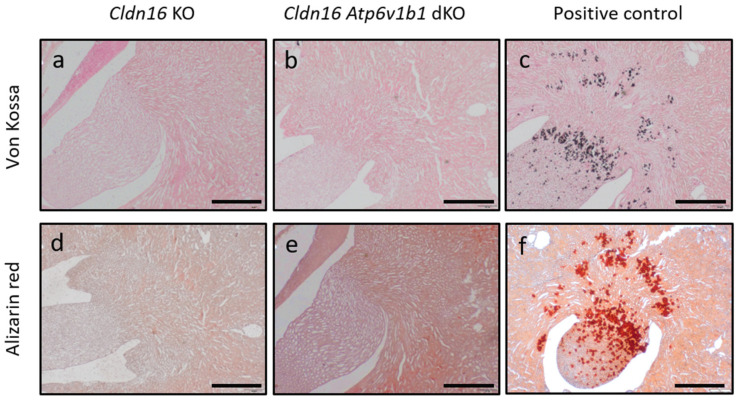
Nephrocalcinosis staining in *Cldn16* KO and *Cldn16 Atp6v1b1* dKO kidney sections. von Kossa staining in kidney sections of *Cldn16* KO (**a**) and *Cldn16 Atp6v1b1* dKO (**b**); Alizarin Red S staining in *Cldn16* KO (**d**) and *Cldn16 Atp6v1b1* dKO (**e**); Von Kossa staining in kidney sections of *Cldn10 ksp* cKO mouse model [19] is shown as a positive control (**c**); Alizarin Red S staining in *Cldn10 ksp* cKO kidney sections is shown as a positive control (**f**); scale bar: 500 µm; n = 3.

## Data Availability

Data are contained within the article. The raw datasets can be requested from the corresponding author upon reasonable request.

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
