# Peer review of "Urinary Acidification Does Not Explain the Absence of Nephrocalcinosis in a Mouse Model of Familial Hypomagnesaemia with Hypercalciuria and Nephrocalcinosis (FHHNC)"

_ijms, 2024, doi:10.3390/ijms25031779_

Round 1

Reviewer 1 Report

Comments and Suggestions for Authors

This is an interesting report that contributes to our understanding the mechanisms of nephrocalcinosis. The authors tested a hypothesis that urine acidification limits formation of calcium deposits in Claudin 16 knockout mice. For that they generated a new transgenic model with dual knockout of Claudin-15 and vesicular ATPase. Expectedly, the dual knockout mice had alkalinized urine. However, despite of such urine alkalinization, the dual knockout animals did not develop nephrocalcinosis. This disproves the acidification hypothesis.

Overall, the report is well written and provides valuable and clinically relevant information. I have just one concern about this paper.

Commets:

1. The authors need to show more clearly loss of Atp6v1b1 expression in kidneys of the dual knockout mice. Immunolabeling presented in Figure 2 is not convincing due to high background staining of the tissue. No primary antibody control should be added. Furthermore RT-PCR analysis showing loss of mRNA expression for Atp6 in the dual knockout animals would be very helpful.

Reviewer 2 Report

Comments and Suggestions for Authors

This study design to generate Atp6v1b1 KO and cross with Cldn16 KO to generate double knockout seems a valid system. However, from the Figure 4 as shown by the study results does not clearly claim what the authors conclude. Below is the reason :

1) The generation of these animals are by CRISPR-Cas9 system and therefore there is a high chance of non-specific genomic integration and therefore it is important to validate the knockouts using atleast three different KO lines. Also, these should be back-crossed with their WT background for atleast three generations before starting to work with the KO for generations of any data.  Only if all the three lines show similar phenotypes the knockouts can be considered homogenous.

2) The Figure 4, legends are not correct what is the legend for panel "f". Also, the Figure 4 panel "b" shows some spots of Von Kossa staining and panel "e" shows increased Alizarin red staining which is opposite to what the authors claim that increasing pH does not affect rather it does affect in the representative images. Also, there is no quantification graphs to show how many times this was repeated, and no statistics analysis is shown. 

3) This study is very minimalistic and not very conclusive. The authors will need to do more work to establish their KO as described in point 1 and validate the lines for the phenotypes. 

Round 2

Reviewer 2 Report

Comments and Suggestions for Authors

The response is satisfactory, and the revisions are appropriate.

Author Response

We sincerely appreciate the effort you made to review our manuscript.